
# Intermittency analysis of charged particles generated in Xe-Xe collisions at $\sqrt{s_{\mathrm{NN}}} = 5.44$ TeV using the AMPT model

Zarina Banoo[⋆] and Ramni Gupta

Department of Physics, University of Jammu, India

⋆ zarina.banoo@cern.ch

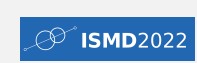

## Abstract

The multiplicity fluctuations are sensitive to QCD phase transition and to the presence of critical point in QCD phase diagram. At critical point a system undergoing phase transition is characterized by large fluctuations in the observables which is an important tool to understand the dynamics of particle production in heavy-ion interactions and phase changes. Multiplicity fluctuations of produced particles is an important observable to characterize the evolving system. Using scaling exponent obtained from the normalized factorial moments of the number of charged hadrons in the two dimensional $(\eta, \phi)$ phase space, one can learn about the dynamics of system created in these collisions. Events generated using Xe-Xe collisions at $\sqrt{s_{\mathrm{NN}}} = 5.44$ TeV with string-melting (SM) version of the AMPT model are analyzed and the scaling exponent ($\nu$) for various $p_T$ intervals is determined. It is observed that the calculated value of $\nu$ is larger than the universal value 1.304, as is obtained from Ginzburg-Landau theory for second order phase transition. Here we will also present the results of the dependence of the scaling exponent on the transverse momentum bin width.

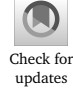

## 1 Introduction

One of the major goals of high-energy heavy ion collision experiments is to understand the particle production mechanism [1]. The spatial fluctuations of the produced particles in the multiparticle production is one of the signatures of criticality and helps to characterize the quark-hadron phase transition [2]. The spatial fluctuations are present at all stages of the collision process, from thermalization, hydrodynamical expansion, hadronization to freeze out. Therefore to collect information about the critical end point and particle production mechanism, it is important to study these types of fluctuations, which are arising out of some dynamical processes [3]. To study the particle density fluctuations, the method of moment was found to be the most suitable one. But in the multiparticle production, the number of hadrons

produced in a single collision is quite small and subject to considerable noise [4]. However the presence of such statistical noise may create problem when the scale dependence of fluctuations is investigated. In that case the use of normalized factorial moments of the multiplicity distribution in a given phase-space volume which filters out the fluctuations of purely statistical origin from the dynamical fluctuations is made [5]. Here we present the results from the study of spatial fluctuations in the charged particles generated in Xe-Xe collisions at $\sqrt{s_{NN}} = 5.44$ TeV using the string melting mode of the AMPT model (version: v1.26t9b-v2.26t9b).

## 2 The Method

To calculate the particle density fluctuations in spatial distributions, the normalized factorial moments are calculated. For this the angular phase space pseudorapidity and azimuthal angle $(\eta, \phi)$ is partitioned into M x M cells. For any value of the number of bins $M$, let $n_i$ be the number of charged particles in a cell. Then the normalized factorial moments $F_q(M)$ of order q is defined by

$$F_q^e(M) = \frac{\left\langle \frac{1}{M^D} \sum_{e=1}^{M^D} n_i(n_i - 1).....(n_i - q + 1) \right\rangle}{\left\langle \frac{1}{M^D} \sum_{e=1}^{M^D} n_i \right\rangle^q}. \tag{1}$$

Here, $\langle \ldots \rangle$ represents the averaging over all events [2]. The power law behaviour of $F_q$ with M

$$F_q(M) \propto M^{\phi_q}, \tag{2}$$

is termed as *intermittency*. Eq.(2) is here referred as *M-scaling* and is considered as a signature of self-similar patterns of particle multiplicity and $\phi_q$ represents the intermittency indices which determines the strength of the intermittency [3]. It has been observed that for the Ginzburg-Landau(GL) formalism, $F_q(M)$ follows power-law [6] as:

$$F_q \propto F_2^{\beta_q}, \beta = (q-1)^\nu, \tag{3}$$

this is referred to as *F-scaling*. Here $\nu$ is the scaling exponent and $\nu = 1.304$ is the universal value given by GL theory for second order phase transition and is independent of the GL parameters [6].

In the SM version of the AMPT model the excited hadronic strings in the overlap volume are converted into partons along the intervening step of decaying hadrons that would have been produced by the Lund string fragmentation process. A detailed description of the model is available in [7]. In this work a sample of around 300K central SM AMPT events generated with an impact parameter $0 \le b \le 3.5$ fm (corresponding to $0 - 5\%$ centrality) have been analyzed.

## 3 Observations and Results

Charged particles produced in the kinematic region $|\eta| \le 0.8$ with full azimuthal coverage and $p_T \le 2.0$ GeV/c are studied for a range of $p_T$ bins as in [6,8]. Normalized factorial moments $F_q$'s are determined for M = 4 - 82 and q = 2, 3, 4,5 and are studied for their dependence on M (M-scaling). Fig 2(a) shows M-scaling for the $p_T$ bin $0.4 \le p_T \le 0.6$ GeV/c. $F_q$ for q =

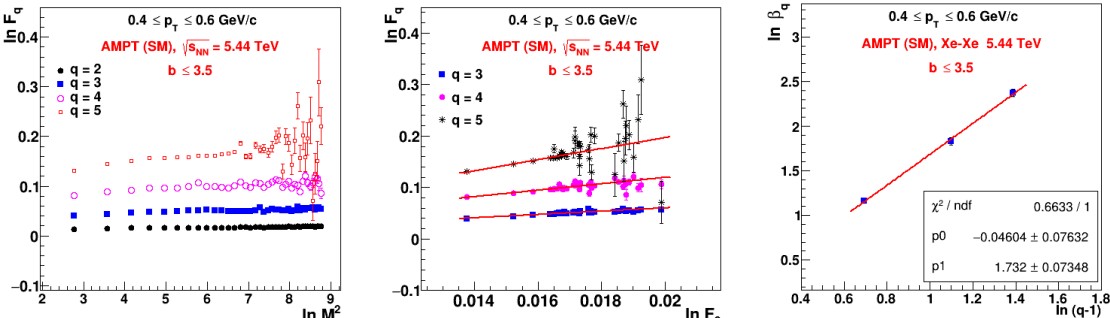

Figure 1: (a) $\ln F_q(M)$ vs. $\ln(M)$. (b) $\ln F_q(M)$ vs. $\ln F_2(M)$ (c) $\ln \beta_q$ vs. $\ln(q-1)$, in two dimensional $(\eta, \phi)$ phase space ($0.4 \le p_T \le 0.6$ GeV/c).

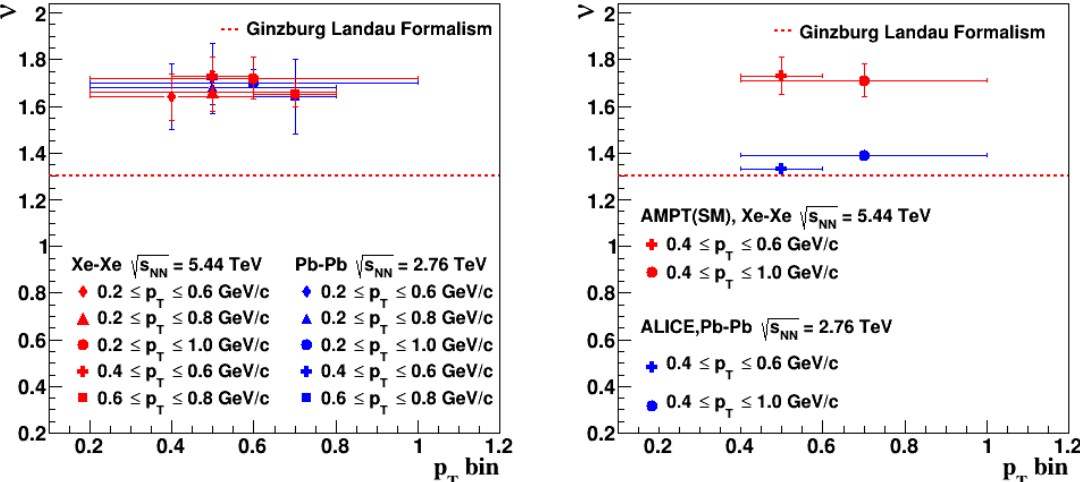

Figure 2: Scaling exponent $\nu$ as a function of various $p_T$ bins for Xe-Xe collisions at $\sqrt{s_{NN}} = 5.44$ TeV and its comparision with Pb-Pb (left) AMPT data [8] and (right) ALICE data [9] .

2, 3, 4 are observed to be independent of M, where for q = 5 at high M large fluctuations in the value of $F_q$ are observed. Which are due to statistical effects. Thus the charged particles generated in AMPT for Xe-Xe collisions do not follow self-similar behaviour.

Dependence of $F_q(M)$ for q = 3, 4, 5 on $F_2(M)$ for the same case, given in Fig 2(b), shows linear behaviour. So even though M-scaling is absent, F-scaling is observed to be present. The scaling exponent '$\nu$' that gives relation between $\beta_q$ and the order of the moments $q$ as defined in Eq.(3) is determined (Fig 2(c)). For all the $p_T$ bins studied, $\nu$ is observed to have average value $\approx 1.7$(Fig 2).

The scaling exponent obtained for Xe-Xe collision from the SM AMPT model is much larger than the value predicted by GL theory for second order phase transition. The AMPT model does not have physics of phase transition implemented into it. This study confirms it and gives significance of intermittency studies in characterizing systems created in heavy-ion collisions.

## 4 Conclusion

Intermittency analysis of Xe-Xe Monte Carlo events at $\sqrt{s_{NN}} = 5.44$ TeV generated with the AMPT model performed to investigate the scaling behaviours and to determine scaling exponent. Whereas M-scaling is absent in charged particles generated in AMPT, F-scaling is observed to be present with scaling exponent ($\nu$) with values different from that for second order phase transition implemented in GL theory and the experimental value from ALICE data. Intermittency methodology is observed to be suitable to characterize various dynamical systems.

## Acknowledgements

**Funding information** Authors are thankful to RUSA 2.0 grant to University of Jammu by the Minisry of Education, Govt. of India for partial support for computing resources.

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
