# Peer review of "Intermittency analysis of charged particles generated in Xe-Xe~collisions at $\sqrt{s_{\rm{NN}}}$ = 5.44 TeV using the AMPT model"

_SciPost Physics Proceedings, doi:SciPost Phys. Proc. 15, 010 (2024)_

## Round 1 · Referee Report · Anonymous (Referee 1) · 2022-11-16

Strengths

1) The work is described clearly and concisely.

Report

The authors look for signs of phase transitions in Xe-Xe collisions using AMPT model by extracting the scaling exponent from multiplicity fluctuations using normalized factorial moments of the charged hadron multiplicity. The scaling exponent is then compared to its universal value obtained from Ginzburg-Landau theory and ALICE data from Pb-Pb collisions. The authors find a value of the scaling exponent that is inconsistent with the universal value for second order phase transitions. This is natural ,since AMPT model does not have the physics of phase transition built-in. The value of the study is that it ensures that the AMPT model does not accidentally reproduce the universal value, and if experiments observe the universal value of the scaling exponent in Xe-Xe collisions, it is more likely to be a sign of a genuine phase transition, rather than a coincidence (since it cannot be reproduced in models which lack a phase transition). I recommend publication

Requested changes

I have noticed some (very) minor issues that are not related to the scientific content of the paper: 1) Typos: "However the presence of such statistical noise may creates problem" -> "create" "Which are due to Statistical effects." I think the word "statistical" is unnecessarily capitalized here 2) References are a bit inconsistent at the moment. Refs. [1]-[5] have journal entries, and refs. [6]-[8] are a bit ambiguous. If refs [6]-[8] refer to these papers: -[6] https://inspirehep.net/literature/1767622 -[7] https://inspirehep.net/literature/1287071 -[8] https://inspirehep.net/literature/1680035 It would be great if the authors could add the journal information also to these items.

  • validity: good
  • significance: good
  • originality: good
  • clarity: high
  • formatting: excellent
  • grammar: good

Author:  Zarina Banoo  on 2022-11-20  [id 3049]

(in reply to Report 1 on 2022-11-16)
Category:
correction

Thank you very much for the excellent report.
I'll make the modifications and resubmit it.

---

## Editorial Decision

published